



# How aerosols and greenhouse gases influence the diurnal temperature range

Camilla W. Stjern[1], Bjørn H. Samset[1], Olivier Boucher[2], Trond Iversen[3], Jean-François Lamarque[4],

Gunnar Myhre[1], Drew Shindell[5], Toshihiko Takemura[6]

[1]CICERO Center of International Climate Research, Oslo, Norway
[2] Institut Pierre-Simon Laplace, Sorbonne Université / CNRS, Paris, France
[3] Norwegian Meteorological Institute, Oslo, Norway
[4] NCAR/UCAR, Boulder, USA
[5] Nicholas School of the Environment, Duke University, Durham, NC, USA
[6] Kyushu University, Fukuoka, Japan

*Correspondence to*: Camilla W. Stjern, camilla.stjern@cicero.oslo.no

**Abstract.** The diurnal temperature range (DTR), or difference between the maximum and minimum temperature within one day, is one of many climate parameters that affects health, agriculture and society. Understanding how DTR evolves under global warming is therefore crucial. Since physically different drivers of climate change, such as greenhouse gases and aerosols, have distinct influences on global and regional climate, predicting the future evolution of DTR requires knowledge of the effects of individual climate forcers, as well as of the future emissions mix, in particular in high emission regions. Using global climate model simulations from the Precipitation Driver and Response Model Intercomparison Project (PDRMIP), we investigate how idealized changes in the atmospheric levels of a greenhouse gas ($CO_2$) and aerosols (black carbon and sulfate) influence DTR, globally and in selected regions. We find broad geographical patterns of annual mean change that are similar between climate drivers, pointing to a generalized response to global warming which is not defined by the individual forcing agents. Seasonal and regional differences, however, are substantial, which highlights the potential importance of local background conditions and feedbacks. While differences in DTR responses among drivers are minor in Europe and North America, there are distinctly different DTR responses to aerosols and greenhouse gas perturbations over India and China, where present aerosol emissions are particularly high. BC induces substantial reductions in DTR, which we attribute to strong modelled BC-induced cloud responses in these regions.

## 1 Introduction

As the global climate warms (Hartmann et al., 2013), changes are not only observed in the daily mean temperature, but in a variety of parameters relevant to society. One such parameter is the diurnal temperature range (DTR), which is a measure of the difference between the maximum and the minimum temperature over a given day. Variations in the magnitude of the DTR



have been found to influence mortality and morbidity (Cheng et al., 2014; Kim et al., 2016; Lim et al., 2012), parasite infection and transmission (Paaijmans et al., 2010), and crop failure (Hernandez-Barrera et al., 2017; Lobell, 2007). Future changes in the DTR is therefore a potential driver of climate impacts, affecting risk assessments associated with health and agriculture and have serious consequences in vulnerable regions.


Observations show a general reduction in DTR over the twentieth century, typically mediated by a stronger increase in the daily minimum temperature ($T_{min}$) than in the daily maximum temperature ($T_{max}$) (Dai et al., 1999; Karl et al., 1993; Vose et al., 2005). This trend in DTR has been linked to anthropogenic emissions, but whether greenhouse gases or aerosols are the dominating influence, and what roles these respective climate drivers will play to future DTR changes, is not clear.

Physically, a range of geophysical processes contribute to determining the land surface DTR of a given region. Maximum temperatures are reached during daytime, due to the excess of incoming shortwave (SW, or solar) radiation. Minimum temperatures occur at night, primarily due to cooling from longwave (LW, or heat) radiation. LW cooling is however active during both daytime and night-time, and thus influences both $T_{min}$ and $T_{max}$, reducing the potential DTR influence of factors affecting it. Thus, greenhouse gases such as $CO_2$, which have a particularly strong effect on LW radiation fluxes throughout

the day (e.g., Lagouarde and Brunet, 1993), are not initially expected to have the strongest direct radiative influence on DTR. Indeed, Dai et al. (1999) showed that changes in water vapor had a relatively small effect on DTR.

Aerosols, on the other hand, primarily have climate interactions affecting the shortwave (SW) spectrum, lowering the incoming SW radiation at the surface, initially reducing the daytime $T_{max}$ and thus reducing DTR. But in addition to the direct interactions with SW and, to a lesser extent, LW radiation, greenhouse gases and aerosols alike have a range of indirect (radiative and non-

radiative) influences on climate, that can cause further changes to $T_{min}$ and $T_{max}$. For instance, sulfate aerosols can interact microphysically with clouds to make them more reflective (Twomey, 1974) or increase the general cloud cover by increasing cloud lifetime (Albrecht, 1989). Cloud changes have been shown to have a strong influence on DTR, primarily by blocking SW radiation and hence reducing $T_{max}$ (e.g., Dai et al., 1999). Increased cloud thickness or cloud cover will also affect the surface energy budget, through increasing downwelling LW radiation. This effect operates during both day and night, but at

high latitudes during the polar night, when there is no incoming SW radiation and the $T_{max}$ effect is therefore absent, an increase in $T_{min}$ from altered LW retention can reduce the DTR. The strong SW atmospheric absorption of BC and $CO_2$ can cause rapid adjustments in both cloudiness and precipitation through their influence on atmospheric stability (Hansen et al., 1997; Richardson et al., 2018; Stjern et al., 2017). An increase in precipitation, for instance, may induce changes in soil moisture, which could in turn influence DTR through a reduced $T_{min}$ due to enhanced evaporation (Zhou et al., 2007). Finally, on a longer

time scale, feedback responses following a warming climate can cause changes to DTR via associated changes in cloud cover (Dai et al., 1999), atmospheric circulation changes, precipitation (Karl et al., 1993), soil moisture (Zhou et al., 2007), surface





heat storage capacity (Kleidon and Renner, 2017), land use changes (Mohan and Kandya, 2015), and the turbulent fluxes of sensible and latent heat in the atmospheric boundary layer (Davy et al., 2017).

Over the coming decades, we can expect to see changes in emissions of both greenhouse gases and aerosols, resulting in a
global backdrop of increased greenhouse gas induced forcing, combined with an aerosol influence that has regionally heterogeneous pattern and potentially strong trends. As an example, the global burden of aerosol loading has recently shifted from Europe to Asia (Myhre et al., 2017a), which has previously been linked to an ongoing drying of the Mediterranean region (Tang et al., 2017), and changes to the South and East Asian Monsoon circulations(Wilcox et al., 2020). However, the future balance between the different climate forcers is highly uncertain, and differs markedly between informed projections such as
the Shared Socioeconomic Pathways (Lund et al., 2019; Rao et al., 2017)

Understanding the separate influence of the different climate drivers on DTR, when taking into account both direct and indirect effects and their climate feedbacks, is therefore an important prerequisite for understanding how regional DTR will evolve over the coming decades. The purpose of this work is to contribute to such an understanding, based on a sample of common, idealized experiments performed by nine coupled climate models. While model studies investigating effects of greenhouse
gases and aerosols on DTR have typically used historical simulations (Lewis and Karoly, 2013; Liu et al., 2016), these simulations include trends in greenhouse gases as well as trends in both scattering and absorbing aerosols, with opposite effects on global mean temperature and, possibly, on DTR. To disentangle the role of different climate drivers to the DTR changes, model responses to idealized experiments where individual drivers are perturbed separately provide a separate line of evidence.

In the present study we compare idealized instantaneous perturbations of $CO_2$, BC and $SO_4$ in nine global climate models from
the Precipitation Driver Response Model Intercomparison Project (PDRMIP) (Myhre et al., 2017b). This unique data sets allows us to investigate whether differing changes to DTR can be expected from trends in greenhouse gases, sulfate or black carbon, and can shed light on results from more comprehensive, multi-forcer simulations, such as those in the Coupled Model Intercomparison Project Phase 6 (CMIP6) (Eyring et al., 2016). While the size of the dataset precludes detailed process-level investigations of the output from each model, any significant changes found based on the median response of the model sample
should represent physically robust expectations based on the geophysical understanding underlying the generation of climate models participating here (which are mostly similar to their CMIP5 configurations; Myhre et al. (2017b)).

In the next section, we give a brief overview of data and methods used in this paper. Section 3 describes the main results of this study, starting with a comparison between PDRMIP baseline DTR values to observations, to show how the specific PDRMIP models capture regional DTR.  The results are summarized in Section 4.






## 2 Methods

We utilize data from the Precipitation Driver and Response Multimodel Intercomaprison Project (PDRMIP), in which nine global climate models have performed idealized simulations of instantaneous perturbations in different climate drivers. Here, we analyze the experiments involving a doubling of $CO_2$ (CO2x2), a tenfold increase in black carbon (BC) (BCx10) and a fivefold increase in sulfate ($SO_4$) (SO4x5), see Table 1. See Figure 1 for the geographical distribution of the perturbed BC and $SO_4$ aerosol concentration fields. The perturbations in the experiments were designed to produce clear and robust climate signals, and the magnitude of the resulting changes are therefore larger than what can be expected from current changes in climate drivers. Moreover, near-equilibrium changes from these abrupt perturbations will likely yield different responses to the gradual build-up (of e.g. $CO_2$) seen in the real word. Note that for the aerosol perturbations, some models perturbed concentrations while others perturbed emissions. This leads to some additional inter-model differences in forcing and response patterns, but has previously been shown not to be a major determining factor for PDRMIP results based on global perturbations (Stjern et al., 2017).

The perturbation experiments are performed and compared relative to baseline simulations representing present-day conditions and using emissions/concentration and solar constant values for year 2000 (except HadGEM2, which used a preindustrial baseline). See Table 1 and (Myhre et al., 2017b; Samset et al., 2016; Stjern et al., 2017) for details and a list of models. All the simulations were 100 years long. Data for the simulation years 51-100 were used in the analyses, and changes were defined as the average of these years for a perturbed simulation minus the corresponding average for the baseline simulation.

DTR was calculated based on daily minimum temperature ($T_{min}$) and maximum temperature ($T_{max}$) values and averaged into monthly and seasonal means. To determine whether a given DTR change is significantly different from zero, regional mean monthly mean DTR values over a 50-year period, for perturbed versus baseline climates, were tested for each model and experiment using Student's t-test ($p < 0.05$).

We have chosen to limit our analysis to land regions and will present results for all land (LND), the United States region (USA), central Europe (EUR), India (IND), eastern China (CHI), and the Arctic (ARC). Regions are chosen partly to present results for the world's most populated regions, and partly where previous findings point to large historical changes in DTR and where changes in the future are of particular interest.

## 3 Results and Discussion

This section presents the global, annual land mean modelled DTR changes in response to the PDRMIP perturbations, as well as regionally and seasonally resolved results. However, as earlier work has demonstrated a tendency in CMIP5-generation models to underestimate DTR relative to observations, with a bias that differs strongly between models and regions (Sillmann et al., 2013), we start our analysis by comparing the PDRMIP baseline DTR values to surface temperature observations.



### 3.1 Comparison to observations

Figure 2 compares PDRMIP DTR, $T_{min}$ and $T_{max}$ for the baseline (year 2000) simulations to gridded observational data from the Climate Research Unit (CRU) TS v. 4.03 (Harris et al., 2014) averaged over years 1991-2010. The observational data set as well as all models are regionally averaged at their native grid resolution. While the multi-model median land annual mean

DTR of 10.8 K is smaller than the CRU value of 11.5 K, individual model values have a standard deviation of 2.56 and range from 8.2 to 15.8 K (Fig. 2a). HadGEM3, NCAR-CESM-CAM4 and CanESM2 have consistently high DTR values, while GISS-E2-R, NorESM1-M and NCAR-CESM-CAM5 have the lowest values. (HadGEM2 has been omitted here, since it used a preindustrial baseline.) As the single-realization simulations performed here will be sensitive to the timing of internal variability among model simulations, this will likely cause some of the inter-model differences. However, the model spread is

not sensitive to the exact time period used. As a crude test, we picked out 20-year periods from the 50 years of the baseline simulations, moving 5 years at a time (giving 7 20-year periods within the 50 years of data), and found that inter-model standard deviations of DTR for these periods ranged between 2.555 and 2.564. While this indicates that model differences are more likely related to actual differences in model formulations and parametrizations, we note that internal variations in regional clouds and precipitation – which strongly influence DTR – can affect trends over periods up to 60 years (Deser et al., 2012),

making it difficult to compare changes in DTR both among models and between models and observations.

Although the geographical DTR pattern is similar between the model median and observations (Fig. 2b), notable differences can also be seen. See for instance western North America, where the modelled DTR is substantially lower than the observed. One known issue in atmospheric models is the representation of the atmospheric boundary layer at high latitudes (e.g., Steeneveld, 2014), where wintertime minimum temperatures are often determined by a very thin and stable boundary layer.

Figure 2a shows that minimum temperatures for most of the models are higher than observations in the northernmost regions investigated here, notably Europe and North America, which explains the underestimated DTR. In the Arctic, however, there is lower model agreement also in estimates of $T_{min}$, with about half the models showing a warm bias, and the other half a cold bias of $T_{min}$. In India and China, on the other hand, $T_{min}$ tends to be too cold, although this is balanced by $T_{max}$ also having a cold bias in many of the models.

Inter-model spread is in all regions larger for $T_{max}$ than $T_{min}$, and for $T_{max}$ there is also more model disagreement as to the sign of the bias relative to observations. Note, for instance, that for $T_{max}$ in USA, four models overestimate while four models underestimate.

Overall, the PDRMIP models perform similarly to CMIP5 models in general (Sillmann et al., 2013), with a general underestimation of DTR, but with large differences between models as well as between regions. Although no direct comparison

between historical DTR changes and the idealized simulations in this study will be made, the caveats noted above should be kept in mind in interpretations of the analyses below.




## 3.2 DTR change in response to different forcing mechanisms

Figure 3 shows how the three drivers ($CO_2$, BC and $SO_4$) influence the DTR for the annual mean (large upper panels) and for
the different seasons (small panels). To make the comparison easier between the drivers, the DTR change is divided by the
global, annual mean temperature change for each driver and model, and thus shows how DTR will change for a 1°C surface
warming due to perturbations in the given climate driver. Model median global temperature change and model spread for the
three drivers are 2.6 [1.5 to 3.7] K (CO2x2), 0.7 [0.2 to 1.7] K (BCx10) and -1.65 [-0.9 to -6.6] K (SO4x), respectively (see
Samset et al. (2016) for core analysis of all PDRMIP experiments and models). For $SO_4$, which cools the climate, this
normalization switches the sign of the change and shows in principle the result of a reduced $SO_4$ level, as opposed to the other
drivers. As the tenfold increase in BC, particularly for some of the models, has a weak impact on global temperatures (Stjern
et al., 2017), normalization by these small numbers leads to particularly large normalized DTR changes for the BCx10
experiment. However, as seen by comparing absolute DTR changes for BCx10 in Fig. S2 to those of CO2x2 and SO4x5 (Figs.
S1, and S3), the absolute DTR change for BCx10 is also large in itself: an annual mean model median DTR change of -0.03
K (compared to -0.05 K for CO2x2) is substantial given than the doubling of $CO_2$ causes a four times stronger response in the
global mean temperature.

The geographical patterns of annual mean DTR change are relatively similar between the drivers. All drivers show reduced
DTR at high latitudes (see the Arctic), increased DTR in mid-latitudes (see, e.g. USA and central/southern Europe), increased
DTR over the Amazon and southern Africa, and reduced DTR over northern/central Africa. The small panels in Fig. 3 shows
that the largest seasonal differences in DTR responses are found between summer (JJA) and winter (DJF) for all driver. In the
next sections we will therefore take a closer look at these two seasons.

### 3.2.1 Wintertime DTR responses

All three climate drivers induce a strong reduction in DTR over northern high and mid latitudes in winter (Fig. 3). Figure 4
shows regional, multi-model mean DTR changes for each season and driver. Colored bars indicate high inter-model
consistency, defined as cases where 80% of models with data have changes of the same sign. In wintertime there is a robust
(colored bars for all drivers) reduction in DTR over Europe and the Arctic. Numbers below the bars indicate for how many of
the nine models these changes are statistically significant, and the number is high for both these regions. A similar reduction
is seen over USA, but here there is lower model agreement on the BC-induced DTR reduction.

As shown in Fig. 5, the wintertime DTR reduction in these northern mid and high latitudes is driven by an increase in $T_{min}$ that
is stronger than the increase in $T_{max}$. Previous studies have shown that while the general global warming, instigated for instance
by increased greenhouse gases, can be expected to increase both $T_{min}$ and $T_{max}$, an increase in cloud cover can substantially
dampen the increase in $T_{max}$ (e.g., Dai et al., 1999), resulting in a DTR reduction. We therefore take a closer look at how
greenhouse gases and aerosols influence the cloud cover in these regions.



Tables S1-S6 show correlation coefficients between changes in DTR and changes in related variables (cloud cover, latent and
sensible heat flux, clear-sky and all-sky downwelling SW radiation and all-sky downwelling LW radiation). Here we see that
there are statistically significant negative correlations between cloud amount changes and changes in DTR for all these regions,
confirming that more clouds are associated with lower DTR. Figure 6 and Table S1 shows cloud cover changes for winter and
summer, for the three drivers.

For CO2x2 and SO4x5, we do find a slight increase in cloud cover in the USA, EUR and ARC regions, which would contribute
to the pattern of $T_{max}$ and $T_{min}$ changes seen in Fig. 5. For BCx10, however, we find a reduction in clouds over Europe but
increases over USA and the Arctic. Wintertime changes in Tmin and Tmax for both aerosol experiments for Europe show very
strong differences and thus strong DTR change (Fig. 5 and Fig. 4). Table S3 shows statistically significant correlations between
DTR change and the change in clear-sky downwelling radiation for these two experiments, and for BCx10 the reduction in
this variable is particularly strong (Table S8) – likely enough to dampen $T_{max}$ in spite of the slight reduction in cloud cover.

In the Arctic region (recall that our regional averages only land, not ocean, areas in this study), the lack of incoming solar
radiation in winter means that the increase in $T_{max}$ will be dampened to a lesser degree, and the difference between the changes
in $T_{min}$ and $T_{max}$ will be smaller. This can be seen in Fig. 5, where the wintertime slopes between $T_{min}$ and $T_{max}$ are much weaker
for the ARC region than, e.g. for EUR, manifesting in a weaker DTR change (Fig. 4).

All in all, a prominent wintertime feature in the EUR, USA and ARC regions is a consistency between drivers in terms of
changes to $T_{min}$ and $T_{max}$, ultimately all causing a reduction in DTR. We see, however, that although greenhouse gases and
aerosols influence DTR in the same manner, the underlying processes differ between drivers.

### 3.2.2 Summertime DTR responses

The reduced wintertime DTR in mid-latitudes is contrasted by a strong summertime increase. Europe stands out as the region
with the best inter-model agreement (Fig. 4; all bars are colored), with a clear summertime DTR increase for all three drivers
that stems from a much stronger increase in $T_{max}$ than in $T_{min}$ (Fig. 7). The same can be seen for USA, albeit with less agreement
between models for the $CO_2$ response. In both these regions, all three drivers induce substantial reductions in summertime
cloud cover (Fig. 6), driving the strong increase in $T_{max}$. The link between DTR and cloud changes is supported by strong and
statistically significant correlations between the two (Tables S2 and S3), with corresponding correlations to sensible heat flux
and the amount of downwelling SW radiation, which we expect to increase as the cloud cover diminishes. A reduction in
summertime precipitation in this region (not shown) contributes to the $T_{max}$ enhancement as a drier climate tends to involve
less clouds and a drier surface with less evaporation. These are conditions that lower the night-time temperatures and increase
daytime temperatures, thus contributing to increased DTR. It is well know from observations that the last decades have seen a
marked drying of Europe in the summer (Manabe and Wetherald, 1987; Rowell and Jones, 2006; Vautard et al., 2014; Leduc
et al., 2019), potentially as a result of an expanding Hadley cell (Lau and Kim, 2015) or due to weaker lapse-rate changes over
the Mediterranean region than over northern Europe (Brogli et al., 2019).



Based on observations, Makowski et al. (2008) found a strong increase in European DTR in the period of strong $SO_2$ mitigations in the region, and suggested a causal relationship. Although natural variability and other forcing mechanisms have likely contributed to these trends, the increase in DTR over Europe seen in the SO4x5 experiment (recall the normalization by temperature change, meaning that this experiment corresponds to a $SO_4$ reduction) is consistent with the findings of Makowski
et al. (2008). However, as our $SO_4$ perturbation experiment causes DTR increases that are comparable with what is caused by perturbations of BC and $CO_2$, it seems that the DTR change in Europe is not a driver specific response solely linked to the trends in aerosols, but rather part of a larger response to the general warming of the climate and the resulting large-scale circulation changes.

In the Arctic region, we find differences in summertime DTR response between the drivers. $CO_2$ causes a stronger Arctic
increase in $T_{max}$ than in $T_{min}$ and thus an increased DTR for all models, while BC for most models causes a stronger increase in $T_{min}$ and thus DTR reduction (Fig. 7). The reason is that $CO_2$ induces a reduction in the summertime Arctic cloud cover, consistent with the increase in $T_{max}$, while BC enhances the cloud cover, thus hindering the strong $T_{max}$ increase. Indeed, calculating SW and LW cloud radiative effects (CRE, Fig. 8) as the difference between clear-sky and all-sky top-of-atmosphere radiative fluxes (see, e.g., Dessler and Zelinka, 2015), we see a strong summertime SW cloud radiative cooling over Arctic
land masses for BCx10 (-7.0 $Wm^{-2}K^{-1}$) (much stronger than the LW CRE effect, thus indicating that the change is primarily to low clouds), contrasting a small positive CRE (+0.2 $Wm^{-2}K^{-1}$) for CO2x2.

We have now shown that, in general, responses to greenhouse gases and aerosols have similar effects on DTR in northern mid and high latitudes. Next, we move on to the high aerosol-emission regions of India and China, to illustrate that in regions of high aerosol emissions, the slight differences in how greenhouse gases and aerosols influence DTR will result in much more
prominent differences in DTR change between the drivers.

### 3.2.3 Driver-specific DTR changes over India and China

Near-term changes in aerosols over India and China, as envisioned in the Shared Socioeconomic Pathways (Rao et al. 2017), project either reduced concentrations of BC and $SO_4$ in both regions, increased concentrations of BC and $SO_4$ in India but reductions in China, or increased BC over both regions but a dipole pattern of increase over India but decrease over China
(Samset et al., 2019). Given this uncertainty in future emission trends, understanding the individual responses of DTR to these two aerosol species is of high interest. In our simulations, BC causes strong DTR changes in all regions (Fig. 4), but particularly in India and China where present-day aerosol concentrations (and thus the magnitude of the perturbations) are high (Fig. 1). There is a high level of agreement between models on the sign of the DTR changes, which is striking, as BC-induced climate changes have been shown repeatedly to be associated with higher levels of model disagreement than changes driven by $CO_2$
and $SO_4$ (Richardson et al., 2018; Samset et al., 2016). Contrasting the strong inter-driver consistency in DTR changes in northern mid latitudes, we find the DTR-response of BC to differ more from the other drivers in India and China, where strongly negative BC-induced DTR changes stand out from the other drivers in both seasons.





Changes in aerosol concentrations have been suggested as a cause of the DTR changes in China (Dai et al., 1999; Liu et al., 2004). Here, we find relatively weak correlations between the DTR changes and changes in the BC burden (0.26 and 0.38 in summer in India in DJF and JJA, respectively, and 0.12 and 0.29 in China). Still, correlations between DTR changes and changes in downwelling clear-sky SW radiation (Tables S4 and S5) are strong and significant, at least in India. Interestingly, for both BCx10 and SO4x5, the aerosol perturbations are stronger in China than in India (Fig. 1), and Table S8 shows that the magnitude of the change in downwelling clear-sky SW radiation is also strongest in China. Still, the link between these changes and DTR are strongest in India. We find that in the BASE simulations, India tends towards a slightly drier climate with less precipitation, less surface evaporation, less cloud cover and a stronger sensible heat flux than China (not shown) – properties typically associated with warmer maximum and colder minimum temperatures. India therefore has a higher DTR to begin with (Fig. 2a), and thus a larger potential for change in the DTR.

In winter, the only substantial DTR changes can be seen for BCx10 in the China region, for which the increase in $T_{max}$ is very weak (Fig. 5), likely due to a simulated increase in clouds for this experiment (Fig. 6). The same can be seen in summer, for which DTR reduction in China due to BC also goes down and cloud levels up. In India, most models agree that the increase in summertime $T_{min}$ is stronger than in $T_{max}$, causing reduced DTR for all three drivers. However, this effect is substantially stronger for BC than for $CO_2$ and $SO_4$. Figure 7 shows that the extremely strong DTR reduction for BCx10 over India in summer occurs because $T_{min}$ is slightly enhanced while $T_{max}$ is actually reduced. The reduction in $T_{max}$ is seen for all models but IPSL-CM5A, which is the only model for which cloud cover decreases over India in this season. For the other models, the summertime cloud cover increase from the BCx10 experiment, as clearly seen in Fig. 6, is substantial over India. In particular, there is a strong reduction in the SW CRE over India (Fig. 8), likely responsible for the reduction in summertime $T_{max}$. Oppositely, the increase in summertime $T_{min}$ (nighttime temperatures are influenced only by the LW spectrum) is enhanced by the positive change in LW CRE over India. In fact, regions which have both a negative change in the SW CRE and a positive change in the LW CRE can be recognized as the regions with the strongest reductions in DTR in the BCx10 JJA map of Fig. 3 (most importantly India and Central Africa).

The strong link between cloud and DTR changes is confirmed by significant negative correlations between DTR and cloud cover in the India and China regions (Tables S4 and S5), strongest in the summer. A previous analysis of the PDRMIP BCx10 experiment by Stjern et al. (2017) found that the BC-induced cloud amount increases in these regions were strongly driven by rapid cloud adjustments (including the so-called semi-direct effect), but were also a part of the longer-term response to increased global surface temperatures. The cloud increases were stronger in India than in China, particularly for low clouds, which have the strongest influence on $T_{max}$.

All in all, while we do see that aerosol-radiation interactions have likely contributed to the regions' DTR changes (through reduction in downwelling SW radiation and thus surface heating), the strongest link again seems to be clouds. Greenhouse gases and aerosols cause distinctly different responses in DTR in the regions – not primarily through their direct radiative





effect, but via their specific influence on cloud cover. As cloud responses to the strong BC perturbations are so substantial, especially in India, the BC response in DTR stands out here.

Given the strong role of clouds in the DTR response, estimates of DTR change will be sensitive to the way that specific climate forcers influence clouds in different climate models, and to their baseline cloud representations. Model responses to $CO_2$ perturbations have been shown to vary greatly between individual models, and responses to aerosols have even larger

uncertainties, partly due to additional variations in parametrizations of indirect and semidirect effects. For instance, both a previous PDRMIP analysis of the BCx10 experiment (Stjern et al., 2017), and an idealized single-model study (Samset and Myhre 2015), indicate that increased BC concentrations lead to rapid adjustments in the form of increased fractions of low clouds and reduced fractions of high clouds over large areas of the globe, with a global mean cooling effect. In a recent study, however, Allen et al. (2019) find indications that the heating rate induced by BC is less "top heavy" than what is calculated in

many climate models (i.e., the vertical profile of short wave heating rates is too uniform), and if the overestimated upper-level cloud response is corrected for, it could instead produce rapid adjustments that warm the climate, on average. These nuances are relevant to the accuracy of DTR simulations as a BC-induced reduction in high clouds will cause LW cooling and likely lower $T_{min}$, while increased low clouds will cause SW cooling and also lower $T_{max}$, with effects on the DTR depending on which is influenced the most. If, on the other hand, BC causes strong reductions in low clouds (increases $T_{max}$) and also weak

reductions in high clouds (reduces $T_{min}$ slightly), this will contribute to an increase in DTR. More research is needed on modelled cloud responses and the vertical distribution on BC, but we note that both Stjern et al. (2017) and Allen et al. (2019) find that in the high-emission regions of India, China and North/Central Africa, the rapid adjustments produce an increase throughout all cloud layers with a total cooling effect (compare to Fig. 8, where the SW CRE is stronger than the LW CRE in these regions) and likely with similar effects on the DTR.

**4 Summary and Conclusion**

We have analyzed a multi-model set of idealized simulations to investigate how changes to the atmospheric levels of $CO_2$, BC and $SO_4$ influence the diurnal temperature range, through alterations of global mean surface temperature, cloud amounts and other climate parameters. For northern mid-latitude regions, we find DTR changes that are broadly similar between drivers. The cause of the DTR change, as apparent from patterns of $T_{min}$ and $T_{max}$ changes, is not always the same for all drivers.

However, the resulting change is consistently an increase in DTR in summer, in EUR, USA and ARC, and a decrease in winter. This similarity may partly be the result of general atmospheric response to changes in surface temperature, rather than the distinct processes through which the drivers operate. Thus, while the strong DTR reductions over Europe have been linked to the massive mitigation effort of $SO_4$ over the past decades, our similar responses of $SO_4$ perturbations to perturbations of $CO_2$ and BC indicate that this is not necessarily an aerosol-specific response.





Over India and China there is less agreement between drivers, with BC causing a strong DTR reduction in both regions in all seasons. The inter-model spread is large, but all models agree on the sign of this change. Although the strong short-wave atmospheric absorption induced by BC particles is predominantly active in daytime, thus impacting the maximum (daytime) temperature more than the minimum (nighttime) temperature, we find that the direct aerosol effect is likely not the leading cause of the DTR response. Rather, it is the strong cloud response to BC in these regions, shown in previous studies to result

from aerosol-induced changes to atmospheric stability and relative humidity, that drive the response in DTR. All models have stronger correlations to cloud related variables than to clear-sky radiative fluxes or changes in BC burden. Hence, the very high BC concentrations in this region have a strong influence on clouds, and thus on DTR.

Although these high-emission regions seem to have driver-specific responses in the DTR, in some seasons, e.g. during autumn over India, $CO_2$ and $SO_4$ produce DTR-changes of the same sign as BC, again indicating the existence of an underlying, driver-

independent DTR response tied to the general warming of the climate. This supports the work of Vinnarasi et al. (2017), who stressed that observed DTR changes over India are a result of both local and global factors working in tandem.

Disentangling the role of aerosols and greenhouse gases to DTR changes is a crucial step towards prediction of future changes in regional DTR. As noted by Vinnarasi et al. (2017), detailed analyses of DTR over regions such as India and East Asia is crucial given the associated risks, which are aggravated by agriculture-dependent economies and dense populations. Moreover,

in these regions, future trends in aerosol emissions are likely to be strong but are also highly uncertain. Understanding how greenhouse gases, absorbing aerosols and scattering aerosols individually influence the DTR may help these regions prepare for future changes.

**Data availability**

The PDRMIP model output is publicly available; for data access, visit http://www.cicero.uio.no/en/PDRMIP/PDRMIP-data-access.

**Author contribution**

CWS, BHS and GM designed the analyses, and CWS carried them out. BHS, OB, JFL and TT performed model simulations.
CWS prepared the manuscript with contributions from all co-authors.

**Competing interests**

The authors declare that they have no conflict of interest.

**Acknowledgements**


PDRMIP is partly funded through the Norwegian Research Council project NAPEX (project number 229778). CWS and BHS were funded through the Norwegian Research Council project NetBC (project number 244141). T. T. was supported by JSPS KAKENHI Grant Number JP19H05669. O.B. acknowledges HPC resources from TGCC under the gencmip6 allocation provided by GENCI (Grand Equipement National de Calcul Intensif).



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








**Tables and Figures**


**Table 1: Overview of models and experiments**

| Experiments | | |
|---|---|---|
| BASE | Present-day conditions, with solar constant and $CO_2$ emissions for year 2000 (Lamarque et al., 2010). Five models ran the aerosol simulations in concentration-based mode, where BC or $SO_4$ concentrations were fixed at the monthly multi-model mean present-day concentrations from AeroCom Phase II (Myhre et al., 2013; Samset et al., 2013). The remaining models (indicated below) ran emission-based simulations where the BASE simulation used present-day emissions of BC or $SO_4$. | |
| CO2x2 | A global instantaneous doubling of the BASE $CO_2$ emissions. | |
| BCx10 | A global instantaneous tenfold increase in the BASE BC concentrations (for the concentration-based models) or emissions (for the emission-based models). | |
| SO4x5 | Like BCx10, only for $SO_4$. For models doing emission-based perturbations, $SO_2$ (not $SO_4$) was perturbed. | |
| **Models** | *Aerosol simulation type* | *No. of lon x lat x lev grid cells* |
| CanESM2 | Emission-based | 128 x 68 x 22 |
| NCAR-CESM1-CAM4 | Concentration-based | 144 x 96 x 17 |
| NCAR-CESM1-CAM5 | Emission-based | 144 x 96 x 17 |
| GISS-E2-R | Concentration-based | 144 x 90 x 40 |
| HadGEM2 | Emission-based | 192 x 144 x 17 |
| HadGEM3 | Concentration-based | 192 x 144 x 17 |
| IPSL-CM5A | Concentration-based | 96 x 96 x 39 |
| NorESM1 | Concentration-based | 144 x 96 x 26 |
| MIROC-SPRINTARS | Emission-based | 256 x 128 x 40 |






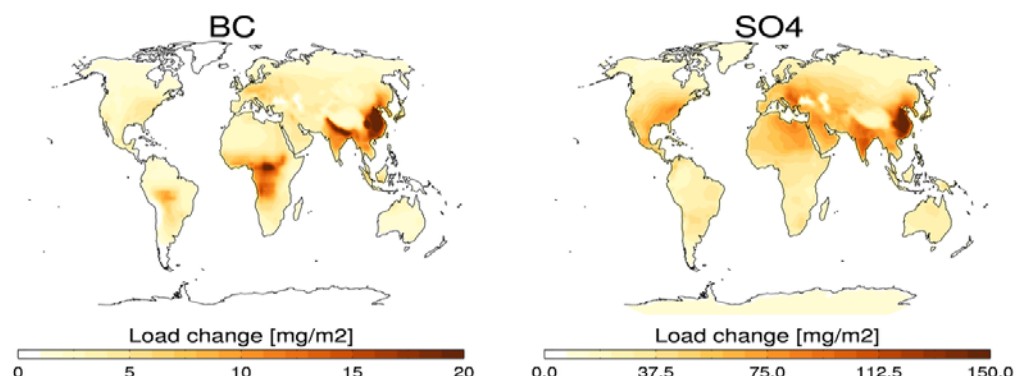

**Figure 1: Geographical distribution of baseline concentrations of BC and SO4, respectively.**



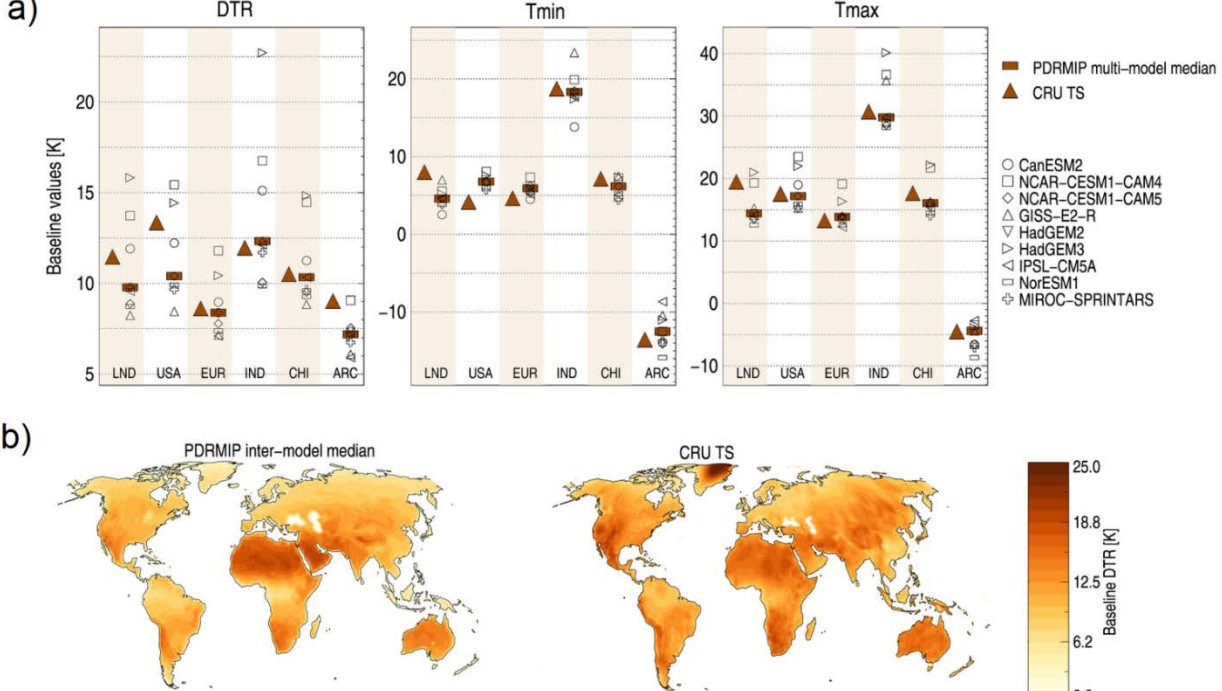

**Figure 2: a) Comparison of baseline (year 2000, years 51-100 of 100-year fully coupled simulations) PDRMIP and CRU TS (mean of years 1991-2010) DTR, Tmin and Tmax. For PDRMIP, single models are shown as open symbols and multi-model median as a**
**filled horizontal bar. Note that as HadGEM2 has a preindustrial baseline in the PDRMIP simulations (Samset et al., 2016) we have omitted this model here. b) Geographical distribution of DTR for the same years, for the PDRMIP inter-model median and for CRU.**







**Figure 3: Multi-model median change in DTR, normalized by the global mean temperature change [K/K], for the three experiments. Large upper maps show annual mean changes, while smaller maps show seasonal changes. Hatching indicates areas where less than 75% of the models agree on the sign of the change. Annual maps include indications of the focus regions of this study. The region called "LND" throughout the manuscript is the average of all land regions on the globe.**

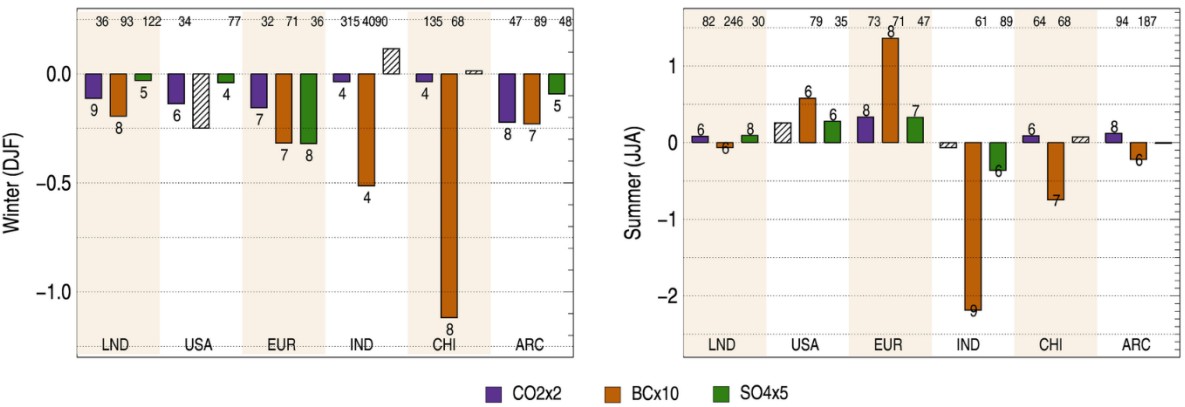

**Figure 4: Multi-model median change in DTR for the different drivers and seasons, normalized by the global mean temperature change [K/K]. Cases for which 80 % of models with data have DTR changes of the same sign are marked with colors, whereas hatched bars indicate larger model disagreement. The numbers associated with the colored bars shows the number of models for which the change is statistically significant (Student's t-test p-value of less than 0.05). The coefficient of variation [std.dev/mean, %] is shown as numbers on the top.**






**Figure 5: Regional wintertime changes in DTR, Tmin and Tmax for the three cases (columns) and six regions (rows). Black horizontal line and squares shows the multi-model median changes.**





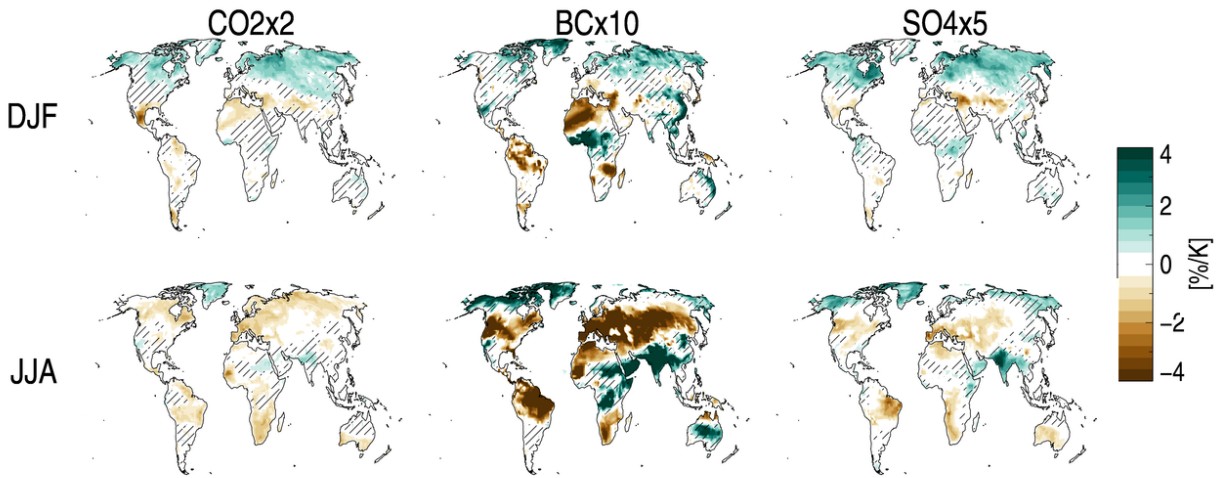


**Figure 6: Multi-model median seasonal cloud cover change for the three drivers, normalized by the global annual mean temperature change. Hatching indicates that less than 75% of the models agree on the sign of the change.**







**Figure 7: Regional summertime changes in DTR, Tmin and Tmax for the three cases (columns) and six regions (rows). Black horizontal line and squares shows the multi-model median changes.**




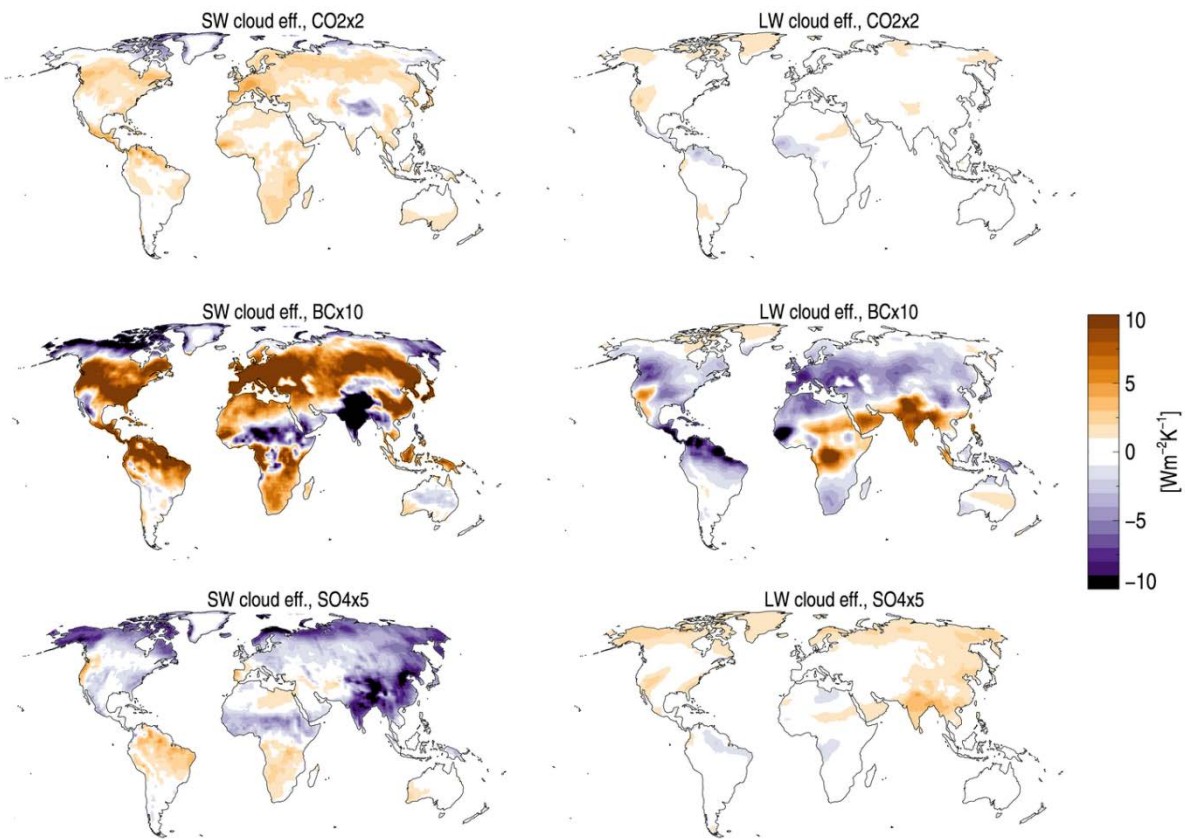

**Figure 8: Multi-model median change in short-wave (SW) and long-wave (LW) cloud radiative effects [Wm⁻²] for the JJA months, for the BCx10 experiment. See supplementary figures for maps of all seasons and experiments.**
