# Peer review of "How aerosols and greenhouse gases influence the diurnal temperature range"

_Atmospheric Chemistry and Physics, 2020_

## Referee Comment (RC1) · Anonymous Referee #2 · 31 May 2020

The paper uses data from Precipitation Driver and Response Model Intercomparison Project (PDRMIP) to investigate how different climate drivers influence changes in diurnal temperature range (DTR) seasonally and regionally. The study is on a topic of broad interest and high relevance. I have a couple of general and specific comments and questions outlined below. Until these points are dealt with I do not think the paper is ready to be published in ACP.

General comments:

1. My main issue with this paper is the number of maps and panels in the figures that are never commented on in the manuscript. My philosophy is that if a figure is not commented on it can be removed. Many sections can be improved by discussing more of what is presented in the figures.

[Figure]

2. The language of the paper can be improved as it contains many very long sentences that makes it difficult to read. It might help to have a native English speaker look over the text.

Specific comments:

3. Figure 1: The caption states this is the baseline concentrations of BC and SO4, while the text (line 95) and figure itself states it shows the perturbed aerosol concentration. Is this figure made based on the models that had concentration perturbations, and not emission? Please give a short explanation of how this will give inter-model differences, and make it clearer in the figure caption what is actually shown.

4. Figure 2: a) The explanation of what years are shown for the models are difficult to read. b) the comma after "same years" make it slightly confusing if you mean the same years as a) CRU or models. Please make even clearer.

5. Figure 3: The caption states that the region called "LND" in the manuscript is average of all land regions, this should not be a figure text but might belong in the methods section instead, where the regions are presented.

6. The author states the regions chosen for investigating were selected based on populated areas, previous findings regarding DTR, and areas where future interest is large. The introduction does not state any regions where previous findings point to large changes in DTR, only regions where anthropogenic aerosol emissions were large (the shift from Europe to Asia). I suggest to either include regions with historically large changes in DTR in the introduction which you can refer to in the methods section, and/or include that regions were chosen based on areas of large anthropogenic emissions of aerosols, as this becomes very important later in the paper.

7. In section 3.1 first paragraph the crude test in inter-model variability might belong in the method-section, or at least before you introduce figure 2. As it stands now it seems unnaturally placed between analysis of Figure 2. Also the introduction of CRU and the

averaging method regarding observations and model grid resolution should have been presented in the methods section rather than results.

8. In section 3.2 the first paragraph states that the perturbation results have been normalized to the temperature change per experiment, which makes both sulfate and BC harder to interpret. I am therefore unsure that this normalization is useful. Please convince the reader why the pros of the normalization outweighs the cons.

9. Figure 3 contains a lot of information that is hardly mentioned in the text. I suggest that if a map is shown but not mentioned in the manuscript it can be moved to supplementary.

10. Please prepare the reader for that China and India is not included in section 3.2.1 and 3.2.2 - analysis of winter- and summer time DTR responses. write clearly that they will be analysed in a later section.

11. The first time you mention China and India as high aerosol emission regions are in the last paragraph of section 3.2.2. This is important information that should have been presented in the introduction.

Minor comments:

line 54: "This effect.." please rewrite this sentence as it is hard to read as it stands.

line 67: The linking of mediterranean drying is unclear if is presented as a cause or effect of the shift in emission. Please rewrite more clearly.

line 75: "these simulations include..", do you mean "these" as the ones in this paper or the typically historical ones? Please rewrite more clearly.

line 97: Do you mean "current" as in present day? Please write more clearly.

Line 125: Standard deviations has the same unit as the original data. Add K throughout the text when standard deviations are presented.

line 138: The text states that atmospheric models misrepresent atmospheric boundary layer in the arctic, and line 141-142 states that the models of this paper have lower agreement in the Arctic than for other regions. Are these two statements related and how? The first statement relate to model/observation comparison, but the second to model/model comparison. Please make clearer in the text.

line 143: The text stated "T_min tends to be too cold in China and India". Please state clearly if you are referring to the "T_min model mean". It reads to me from the figure that model mean T_min in India is not much colder than CRU.

line 145: A specific point is made about USA having four models overestimating and four models underestimating T_max, as USA does not differ largely from the other regions in Figure 2a) for T_max this example is not needed. The inter-model spread is well presented in Figure 2a.

Line 158: to make it easier to read maybe write on the format "2.6 [1.5 to 3.7] K for CO2x2" instead of parentheses. Also the number 5 is missing in the perturbation representation for sulfate (SO4x5).

line 243: Please state clearly what figure you are referring to with this statement.

line 249-250: what are the units of the number in parenthesis?

line 271: should say cloud cover - not only cloud.

line 273: cloud amount or cloud cover? These are two different metrics.

line 275: "cloud increases" should say cloud amount/cover increases.

line 278: "strongest link" between aerosol-radiation and DTR? please rewrite more clearly what is linked and how.

line 280: "As cloud responses to the strong BC perturbations are so substantial, especially in India, the BC response in DTR stands out here." please rewrite more clearly what you want to say with this sentence.

line 287-291: Divide this long sentence into smaller ones.

line 314: Please add citation for the "previous studies".

line 324: "Moreover,.." please rewrite this sentence, as it is hard to read as it stands.

―――――――――――――――――

---

## Referee Comment (RC2) · Anonymous Referee #1 · 23 Jun 2020

This paper presents an analysis of the PDRMIP experiments and assesses the effect of different agents ($CO_2$, black carbon, and sulphate) on the diurnal temperature range. This is an important topic of research and the analysis presented here makes a valuable contribution. However, there are a few minor issues that should be addressed prior to publication.

General comments: I think the figures are useful and well presented, but in a couple of places I miss some more elaboration on the results shown therein. See specific comments below.

Minor points: L45: As you allude to later (L63) the relationship between radiative forcing and SAT is non-linear (especially in shallow, stably-stratified conditions, such as mid-lat winter Tmin) because it is modified by the near-surface mixing strength; I think this

[Figure]

should be clarified here.

L55: In the previous sentences you argue that LW changes effect both the Tmax and Tmin, but that SW changes affect the Tmax more strongly, but then here you should make clear that in the polar night, in the absence of SW, LW changes effect both Tmax and Tmin.

L69: "Informed projections" I think you should expand on what you mean by that i.e. pathways derived from IAMs

L113-114: I think the choice of the Arctic as a region of interest needs some clearer justification as you have already mentioned the DTR here is not so much driven by diurnal variations in SW forcing.

L124: The multi-model median is referred to as 10.8K, but in the corresponding figure 2 this looks like it is less than 10K – am I missing something or is this number referring to the mean perhaps?

L136: Here and elsewhere when you refer to comparison of geographical patterns the analysis is qualitative, but it would benefit from being supplemented by some quantitative measures of pattern similarity e.g. correlation coefficient between patterns.

Related to Figure 2: For the Tmin plot all the individual regions are either warmer or the same temperature in the models as compared to the observations, while in the LND average the models are colder. Since this somewhat undercuts the argument about choosing representative regions around the globe, I think this should be commented on.

L167: Again, we have a qualitative statement about pattern similarity which would benefit from a quantitative statement to support it.

Technical issues: L66: pattern(s)

---

## Author Comment (AC1) · 20 Aug 2020

————Response to Reviewer #1————

>General comments: I think the figures are useful and well presented, but in a couple of places I miss some more elaboration on the results shown therein. See specific comments below.

We thank the reviewer for performing this review and refer to specific comments and their responses in blue below. We have gone through all our figures with you comment in mind, and either elaborated more on what they show, or moved the figure to the supplementary.

>L45: As you allude to later (L63) the relationship between radiative forcing and SAT

is non-linear (especially in shallow, stably-stratified conditions, such as midlat winter Tmin) because it is modified by the near-surface mixing strength; I think this should be clarified here.

We agree that this could have been made clearer. We have rewritten most of the introduction to improve the clarity, and have also added the following sentence: "Finally, each process and its effect on DTR may be modified by non-linear effects such as, e.g., local hydrological conditions or atmospheric stratification."

>L55: In the previous sentences you argue that LW changes effect both the Tmax and Tmin, but that SW changes affect the Tmax more strongly, but then here you should make clear that in the polar night, in the absence of SW, LW changes effect both Tmax and Tmin.

The mention of the polar night in this paragraph was a bit confusing and uncomplete, as noted by the reviewer. We have determined to remove this from the introduction, and rather explain Arctic-related processes when we present the Arctic results.

>L69: "Informed projections" I think you should expand on what you mean by that i.e. pathways derived from IAMs

Thank you, we have reworded this sentence, and it now reads: "However, the future balance between the different climate forcers is highly uncertain, and differs markedly between the various Shared Socioeconomic Pathways currently in use by the projection and climate impact communities (Lund et al., 2019; Rao et al., 2017). In particular, they include a wide range of possible emission combinations of BC and SO4 from India and China, some of which lead to a strong dipole pattern in regional, aerosol induced radiative forcing over the coming decades (Samset et al., 2019)."

>L113-114: I think the choice of the Arctic as a region of interest needs some clearer justification as you have already mentioned the DTR here is not so much driven by diurnal variations in SW forcing.

We did mention in the introduction this insensitivity to SW radiation in the Arctic during the polar night. But in the other half of the year, SW forcing is important even during the night, and thus the driver variation in SW forcing will be highly present in the Arctic during summer. However, in a general rewriting of the Introduction to improve clarity and readability, we have chosen to remove this part. Instead, the final paragraph of the Methods section now contains a sentence motivating the inclusion of the ARC region: "We present results for all land regions aggregated (LND), and the populated, high (present or previous) aerosol emission regions of the continental United States (USA), central Europe (EUR), India (IND), eastern China (CHI). In addition, we study changes in the Arctic (ARC), which is a region known to be sensitive to remote emissions but where the mediating processes are not fully explored. As an example, potential drivers of regional impacts such as melt ponds and sea ice loss may depend on summertime Arctic DTR, , which may in turn depend on diurnal variations in, e.g., photochemical particle production or transport into the region (Deshpande et al., 2014). Our main focus is however on the major aerosol emission regions."

>L124: The multi-model median is referred to as 10.8K, but in the corresponding figure 2 this looks like it is less than 10K – am I missing something or is this number referring to the mean perhaps?

Thank you for spotting this error, it is indeed slightly less than 10 – the number should be 9.8 and not 10.8.

>L136: Here and elsewhere when you refer to comparison of geographical patterns the analysis is qualitative, but it would benefit from being supplemented by some quantitative measures of pattern similarity e.g. correlation coefficient between patterns.

This specific sentence referred to here is now removed. Not supporting statements of geographic similarity with spatial correlation coefficients was a conscious choice on our part. These maps are meant to give the reader an overview of the general DTR changes, and we also wanted to show how the changes look in regions and

seasons other than the ones we focus our analyses on. Still, we want the analyses to be focused on the regional averages, and find that there is no quick way to just add global comparisons such as spatial correlation coefficients. We would then have to supply correlations between BCx10 and CO2x2, between SO4x4 and CO2x2 and between BCx10 and CO2x2, and would have to explain them from a global perspective. This, we feel, is beyond the scope of this manuscript. Still, we understand the reviewer's comment. We have chosen to solve the issue by more careful wording when we point to these maps, and by informing the reader that quantitative results will be supplied. For example, section 3.2.1 now starts with "As visible in Fig. 3, all three climate drivers induce a strong reduction in DTR over northern high and mid latitudes in winter. In Fig. 4 we quantify these changes by taking a closer look at regional averages."

>Related to Figure 2: For the Tmin plot all the individual regions are either warmer or the same temperature in the models as compared to the observations, while in the LND average the models are colder. Since this somewhat undercuts the argument about choosing representative regions around the globe, I think this should be commented on.

This inconsistency turned out to stem from deficient masking – especially coastal and island regions were not included in the PDRMIP data, causing values that in the case of Tmin were too low when averaged over all land. We have now fixed this problem and find that the all-land average Tmin is 7.8 in PDRMIP and 6.4 in CRU-TS. In response to the response from other reviewers, however, we have chosen to change Figure 2 to make it more intuitive. We therefore plot PDRMIP-CRU differences instead of PDRMIP and CRU values separately. The maps (panels b) also show differences now, for both DTR, Tmin and Tmax.

>L167: Again, we have a qualitative statement about pattern similarity which would benefit from a quantitative statement to support it.

Se response to comment on line 136.

>Technical issues: L66: pattern(s)

Thank you, this is now corrected.

—————-Response to Reviewer #2—————

>General comments: >1. My main issue with this paper is the number of maps and panels in the figures that are never commented on in the manuscript. My philosophy is that if a figure is not commented on it can be removed. Many sections can be improved by discussing more of what is presented in the figures.

We thank the reviewer for this input, and we agree that figures should be properly discussed if included in the main manuscript. We have solved this issue partly by making sure to discuss figures more thoroughly, and partly by moving one of the figures into the supplementary. Specifically, the two informationally heavy many-panel figures containing Tmin and Tmax changes (one figure for summer and one for winter) have been grouped by region instead of season, and the USA-EUR-ARC version have been moved to the supplementary.

>2. The language of the paper can be improved as it contains many very long sentences that makes it difficult to read. It might help to have a native English speaker look over the text.

When reading over the manuscript again, we indeed see that the language has potential for improvement. We have done a language "clean-up", with specific focus on dividing and simplifying the long sentences, and hopefully have a manuscript that is more readable now.

>3. Figure 1: The caption states this is the baseline concentrations of BC and SO4, while the text (line 95) and figure itself states it shows the perturbed aerosol concentration. Is this figure made based on the models that had concentration perturbations, and not emission? Please give a short explanation of how this will give inter-model differences, and make it clearer in the figure caption what is actually shown.

The confusion is understandable – we have now corrected the text (it is the baseline concentrations that are shown in the figure). We have also added a few sentences on how the differences in model set-up may induce differences in responses: "Note that for the aerosol perturbations, four of the ten models perturbed concentrations while others, due to variations in model design, used year 2000 emissions as a baseline and perturbed these emissions instead. This leads to some additional inter-model differences in forcing and response patterns. For instance, in concentration-driven simulations, climate dynamics (e.g., a change in precipitation and thus wet deposition) will not influence BC concentrations, while feedbacks between BC and other climate processes can operate in emission-driven simulations. However, a previous PDR-MIP study found the difference between climate responses in emission-driven versus concentration-driven experiments to be highly model dependent (Stjern et al., 2017). At least for the BCx10 simulations, two of the emission-driven models (CESM-CAM5 and MIROC-SPRINTARS) showed responses very similar to the concentration-driven models, while the two others (HadGEM2-ES and CanESM2) had slightly stronger responses that might be related to the nature of the experiment set-up."

>4. Figure 2: a) The explanation of what years are shown for the models are difficult to read. b) the comma after "same years" make it slightly confusing if you mean the same years as a) CRU or models. Please make even clearer.

Thank you, we agree that this caption was difficult to read. Note that we have changed the figure slightly, and the caption now reads: "Figure 2: For DTR, Tmin and Tmax, respectively, the figure shows a) geographical distribution of CRU TS values, averaged over years 1991-210, b) geographical distribution of differences between the PDRMIP model-median baseline (mean of years no. 51-100 of 100-year fully coupled simulations) and CRU TS, and c) regionally averaged differences for the model median and for individual models. Note that as HadGEM2 has a preindustrial baseline in the PDRMIP simulations (Samset et al., 2016) we have omitted this model here."

>5. Figure 3: The caption states that the region called "LND" in the manuscript is

average of all land regions, this should not be a figure text but might belong in the methods section instead, where the regions are presented.

The LND region is indeed defined in the methods sections as well, but we understand that this additional mention of it does not belong in the figure caption. We have now simply removed this sentence from the caption.

>6. The author states the regions chosen for investigating were selected based on populated areas, previous findings regarding DTR, and areas where future interest is large. The introduction does not state any regions where previous findings point to large changes in DTR, only regions where anthropogenic aerosol emissions were large (the shift from Europe to Asia). I suggest to either include regions with historically large changes in DTR in the introduction which you can refer to in the methods section, and/or include that regions were chosen based on areas of large anthropogenic emissions of aerosols, as this becomes very important later in the paper.

Thank you, this is very good advice. In the section following the description of DTR-related processes in the Introduction, we have now clarified that our choice of regions was mainly based on past or present anthropogenic aerosol trends. In addition, the final paragraph of the Methods section repeats the motivation of the choice of regions: "We present results for all land regions aggregated (LND), and the populated, high (present or previous) aerosol emission regions of the continental United States (USA), central Europe (EUR), India (IND), eastern China (CHI). In addition, we study changes in the Arctic (ARC), which is a region known to be sensitive to remote emissions but where the mediating processes are not fully explored."

>7. In section 3.1 first paragraph the crude test in inter-model variability might belong in the method-section, or at least before you introduce figure 2. As it stands now it seems unnaturally placed between analysis of Figure 2. Also the introduction of CRU and the averaging method regarding observations and model grid resolution should have been presented in the methods section rather than results.

We agree with the suggested edits, and have now moved the variability-test, as well as the introduction of CRU into the methods section, improving the flow of Section 3.1.

>8. In section 3.2 the first paragraph states that the perturbation results have been normalized to the temperature change per experiment, which makes both sulfate and BC harder to interpret. I am therefore unsure that this normalization is useful. Please convince the reader why the pros of the normalization outweighs the cons.

We understand the reviewer's objection to this normalization. The designs of the PDR-MIP experiments aim for perturbations of similar climate impact magnitude, but it still feels rather arbitrary to compare a doubling of CO2 to, e.g., a fivefold increase in SO4. We believe results are more readily compared if they all reflect "the response seen if the given component were to warm global climate by one degree." To give the reader a better understanding of why we use this normalization, we have moved this into a separate part of the Methods section, to avoid a lengthy explanation in between presentation of results. That paragraph reads: "Using step perturbations rather than transient simulations means that climate responses will be different to those seen in the real world. The advantage is that signals more rapidly emerge from the noise of internal variability, provided that the forcing applied is of sufficient strength. In PDRMIP, the experiments were designed to produce such clear and robust climate signals. The experiments are however not identical in effective radiative forcing, which necessitates some normalization if the results are to be fully comparable. Here, we have chosen to divide climate responses (e.g., the DTR change) by the global, annual mean temperature change for each driver and model. Our comparisons therefore show the response expected for a $1°C$ surface warming due to perturbations in the given climate driver."

>9. Figure 3 contains a lot of information that is hardly mentioned in the text. I suggest that if a map is shown but not mentioned in the manuscript it can be moved to supplementary.

We agree that some of the figures were not commented on enough to justify inclusion

in the main manuscript. We have now made sure to point to Figure 3 throughout the discussion of the results, to justify its presence.

>10. Please prepare the reader for that China and India is not included in section 3.2.1 and 3.2.2 - analysis of winter- and summer time DTR responses. write clearly that they will be analysed in a later section.

Thank you, a proper introduction of the sections to come is now added at the end of the introductory paragraphs of 3.2: "In the next sections we will therefore take a closer look at these two seasons – first for the high and mid northern latitude regions USA, EUR and ARC, and finally for the Asian regions IND and CHI."

>11. The first time you mention China and India as high aerosol emission regions are in the last paragraph of section 3.2.2. This is important information that should have been presented in the introduction.

Absolutely, this is now fixed by the new paragraph in the introduction, as mentioned above.

>Minor comments: >line 54: "This effect.." please rewrite this sentence as it is hard to read as it stands.

We agree, the sentence was hard to read. The sentence was however removed in its entirety when rewriting the introduction.

>line 67: The linking of mediterranean drying is unclear if is presented as a cause or effect of the shift in emission. Please rewrite more clearly.

We meant to write that the aerosol emission shift was a cause of the drying – the text is now improved.

>line 75: "these simulations include..", do you mean "these" as the ones in this paper or the typically historical ones? Please rewrite more clearly.

A change from "these simulations" to "such simulations" clarifies that we refer to the

historical simulations being discussed in that sentence.

>line 97: Do you mean "current" as in present day? Please write more clearly.

"Current changes" are now changed to "preindustrial to present-day changes".

>Line 125: Standard deviations has the same unit as the original data. Add K throughout the text when standard deviations are presented.

Units for the standard deviations are now added.

>line 138: The text states that atmospheric models misrepresent atmospheric boundary layer in the arctic, and line 141-142 states that the models of this paper have lower agreement in the Arctic than for other regions. Are these two statements related and how? The first statement relate to model/observation comparison, but the second to model/model comparison. Please make clearer in the text.

The comment on the inter-model spread was meant to illustrate that not all models may be bad at representing the atmospheric boundary layer, but this point was not clearly made in the text. We have now rewritten much of this section.

>line 143: The text stated "T_min tends to be too cold in China and India". Please state clearly if you are referring to the "T_min model mean". It reads to me from the figure that model mean T_min in India is not much colder than CRU.

In the observational comparison figure, we found an inconsistency that turned out to stem from deficient masking – especially coastal and island regions were not included in the PDRMIP data. We have now fixed this problem, and slight changes to Fig. 2 as a result meant that we had to delete this statement altogether. Note that we have chosen to change Fig. 2 to make the comparison more intuitive, the main difference being that we show model biases (model-observation differences) instead of absolute values. We hope that this makes the discussion easier to follow.

>line 145: A specific point is made about USA having four models overestimating and

four models underestimating T_max, as USA does not differ largely from the other regions in Figure 2a) for T_max this example is not needed. The inter-model spread is well presented in Figure 2a.

Agreed, this was only meant as an example, but we see that it seems that we believe that this region stands out from the others somehow. The example is now removed.

>Line 158: to make it easier to read maybe write on the format "2.6 [1.5 to 3.7] K for CO2x2" instead of parentheses. Also the number 5 is missing in the perturbation representation for sulfate (SO4x5).

The syntax is now changed (and the missing 5 added), as suggested by the reviewer.

>line 243: Please state clearly what figure you are referring to with this statement.

Thank you, this is clarified now.

>line 249-250: what are the units of the number in parenthesis?

These number were Pearson's correlation coefficients (unitless) – this is clarified now.

>line 271: should say cloud cover - not only cloud.

Thank you, we have now corrected this.

>line 273: cloud amount or cloud cover? These are two different metrics.

Cloud cover – this is now specified, and we have gone through the rest of the document to make sure this is consistent all over.

>line 275: "cloud increases" should say cloud amount/cover increases.

Thank you, it now says "cloud cover increases".

>line 278: "strongest link" between aerosol-radiation and DTR? please rewrite more clearly what is linked and how.

Thank you, we have now changed the wording: "the strongest link seems to be clouds"

→ "the strongest driver of DTR changes- seems to be clouds"

>line 280: "As cloud responses to the strong BC perturbations are so substantial, especially in India, the BC response in DTR stands out here." please rewrite more clearly what you want to say with this sentence.

Thank you, the sentence now reads "As the magnitude of the BC-induced cloud response is particularly strong over India, this is where we see the most substantial DTR reduction".

>line 287-291: Divide this long sentence into smaller ones.

Thank you, both the indicated sentence as well as the previous sentence was divided in two for increased clarity.

>line 314: Please add citation for the "previous studies".

Reference is now added.

>line 324: "Moreover,.." please rewrite this sentence, as it is hard to read as it stands.

Agreed, the final section did not read well, it is now changed to: "Disentangling the role of aerosols and greenhouse gases to DTR changes is a crucial step towards prediction of future changes in regional DTR. This is particularly true in regions such as India and East Asia (Vinnarasi et al., 2017), in which risk factors are aggravated by agriculture-dependent economies and dense populations, and where future trends in aerosol emissions are highly uncertain but likely to be strong. Understanding how greenhouse gases, absorbing aerosols and scattering aerosols individually influence the DTR may help these regions prepare for future changes."